# A Zn(II)-Based Sql Type 2D Coordination Polymer as a Highly Sensitive and Selective Turn-On Fluorescent Probe for Al^3+^

**DOI:** 10.3390/molecules26237392

**Published:** 2021-12-06

**Authors:** Dmitry I. Pavlov, Alexey A. Ryadun, Andrei S. Potapov

**Affiliations:** Nikolaev Institute of Inorganic Chemistry, Siberian Branch of the Russian Academy of Sciences, 3 Lavrentiev Ave., 630090 Novosibirsk, Russia; pavlov@niic.nsc.ru (D.I.P.); ryadunalexey@mail.ru (A.A.R.)

**Keywords:** 2,1,3-benzothiadiazole, 1,2,4-triazole, crystal structure, luminescence, coordination polymer, metal-organic framework, aluminum detection, fluorescence enhancement

## Abstract

A luminescent coordination polymer with the overall formula {[Zn(tr_2_btd)(bpdc)]∙DMF}_n_ (where tr_2_btd = 4,7-di(*1H*-1,2,4-triazol-1-yl)-2,1,3-benzothiadiazole; bpdc = 4,4′-biphenyldicarboxylate) was synthesized and characterized by single-crystal and powder X-ray diffraction, thermogravimetric, infrared spectroscopy, and elemental analyses. Luminescent properties of the obtained compound were studied in detail both in the solid state and as a suspension in N,N-dimethylacetamide (DMA). It was found that {[Zn(tr_2_btd)(bpdc)]∙DMF}_n_ exhibits bright turquoise luminescence with excellent quantum efficiency and demonstrates turn-on fluorescence enhancement effect upon soaking in DMA Al^3+^ solution. Fluorescence titration experiments were carried out and the detection limit for Al^3+^ ions was calculated to be 120 nM, which is among the lowest reported values for similar materials. Moreover, compound demonstrated excellent selectivity and reusability, and the mechanism of the response is discussed. These results indicate that {[Zn(tr_2_btd)(bpdc)]∙DMF}_n_ is a promising probe for sensitive fluorescent Al^3+^ detection.

## 1. Introduction

Detection of metal cations is an important task in a broad range of human activities. Their presence can cause interferences in materials processing, pose an environmental threat and danger to the public health. It is well-known that lead, cadmium, and mercury are extremely toxic to organisms [1,2,3]. Although its effect is not as dramatic as in the case of lead, aluminum can and does present a concern regarding its influence on the human health [4]. Aluminum is a ubiquitous metal, encountered daily in foods, beverages (including drinking water), air-borne particulates and fumes, kitchenware, alloys, pharmaceuticals, vaccines [5]. Due to its wide spread throughout general population’s lives, an extensive research into aluminum physicochemical characteristics in relation to its uptake, buildup, and systemic bioavailability has been carried out [6]. This metal was confirmed as potentially toxic to the skeletal and hematopoietic systems [7,8]. There is an ongoing dispute considering the role of aluminum in the development of several neurodegenerative diseases, such as Alzheimer’s disease or Parkinson’s disease [9]. Several reports have attempted to link aluminum content in drinking water with dementia. In the early 1990s the case has been made that the evidence is strong enough to imply that a significant cutting of aluminum exposure would significantly reduce the prevalence of Alzheimer’s disease [6]. Current exposure rates may be as little as 0.03 µg/kg/day in clean environment to 233 µg/kg/day in polluted environment [8]. Although the particular effect of aluminum exposure on etiology of neurodegenerative diseases is still debated, most reviews agree on several points: (i) Aluminum is very prevalent in the environment and there is some level of human consumption; (ii) neurotoxicity of high aluminum levels is well established; (iii) there is repeated epidemiological correlation between ingested aluminum and incidence of Alzheimer’s disease [9,10]. Thus, it is evident that aluminum exposure should be kept at minimum as its biological effects are not fully understood yet.

Recently, a significant effort has been put into developing alternative methods for the detection of aluminum species since traditional methods such as chromatography [11], potentiometric methods and voltammetry [12], atomic absorption spectrometry [13], atomic emission spectrometry [14] have the disadvantages of complexity, requirement for expensive equipment, and non-real-time detection. Fluorescent sensing overcomes these disadvantages by offering high sensitivity along with selectivity, ease of operation, real-time response, and possibility for naked-eye recognition of the analytical signal [4,15,16,17,18,19].

Metal-organic frameworks (MOFs) also known as porous coordination polymers (PCPs), are porous crystalline highly ordered solids constructed by bridging metal ions or metal clusters with polytopic organic ligands. Their potential applications include catalysis [20,21], separation, and storage of gas and liquid mixtures [22,23,24,25], chemical sensing [26]. Utilization of MOFs as fluorescent probes offers additional advantages, such as improved selectivity and sensitivity, and reusability. In the recent decade, a vast array of metal-organic framework materials capable of selectively detecting metal ions, small molecules, nitroaromatics, etc., were reported. However, most Al^3+^-sensitive MOFs demonstrate luminescence quenching response [27,28,29]. To the best of our knowledge, “turn-on” selective MOF sensors for Al^3+^ are still a relatively rare occurrence [30,31].

Herein, preparation of a new 2,1,3-benzothiadiazole-based fluorescent coordination polymer {[Zn(tr_2_btd)(bpdc)]∙DMF}_n_ (**1**), constructed using 4,7-di(1H-1,2,4-triazol-1-yl)-2,1,3-benzothiadiazole (tr_2_btd, Figure 1), 4,4′-biphenyldicarboxylic acid (H_2_bpdc, Figure 1) as linker is reported. It was found that the coordination polymer **1** exhibits high selectivity and sensitivity toward Al^3+^ via a fluorescence enhancement (turn-on) effect. The determined detection limit of Al^3+^ ions of 120 nM is among the best of the reported Al^3+^ MOF sensors.

## 2. Results and Discussion

### 2.1. Synthesis

The tr_2_btd ligand was synthesized in a moderate yield by the reaction of Br_2_btd with 1H-1,2,4-triazole in a superbasic medium K_3_PO_4_–DMSO (Figure 1), the pure product was obtained by simple recrystallization, which presents an advantage compared to the previously reported method that requires column chromatography [32].

Coordination polymer {[Zn(tr_2_btd)(bpdc)]∙DMF}_n_ (**1**) was conveniently obtained in high yield by reacting tr_2_btd, H_2_bpdc and Zn(NO_3_)_2_∙6H_2_O in a mixed solvent system DMF:EtOH (3:1) at 100 °C.

### 2.2. Structural Characterization

Compound **1** crystallizes in a centrosymmetric monoclinic P2_1_/c space group. The asymmetric unit consists of three separate fragments. Each fragment includes a crystallographically independent zinc ion bonded by one dicarboxylate and one tr_2_btd molecules (Figure 2). There are three DMF molecules per asymmetric unit, two of them participate in hydrogen bonding with 1,2,4-triazole rings (Appendix A), distance d(O16–C33) is 3.108 Å, and d(O4–C57) is 3.157 Å, while the other is disordered over two positions (occupancies 0.6 and 0.4). The coordination sphere of each Zn ion consists of four oxygen atoms from separate bpdc^2−^ anions and two nitrogen atoms from separate tr_2_btd molecules (Figure 2a). The distances d(Zn–O) are in the range of 1.976(2)–2.0322(19) Å for the shorter bond and 2.3929(19)–2.633(3) Å for the longer bond. The distances d(Zn–N) are in the range of 2.033(2)–2.099(2) Å. The coordination polyhedra of Zn^2+^ ions can be treated as distorted octahedrons. Zinc cations are linked by tr_2_btd molecules into polymeric chains which are cross-linked by bpdc^2−^ anions to form 2D layers. Three layers interpenetrate each other to form supramolecular sheets which are packed into stacks held together by weak π–π interactions between tr_2_btd molecules and hydrogen bond between one of the bpdc^2−^ carboxylate oxygen atoms and 1,2,4-triazole ring (Appendix A). As a result of such stacking, channels propagating along the crystallographic axis *a* are formed with the approximate size of 4 × 8 Å. Free void volume of the structure calculated by PLATON software is 24%. In the as-synthesized structure, these channels are partly occupied by DMF molecules, but are potentially capable of including other small molecules and/or ions upon activation. Topological analysis using ToposPro [33] suggests that coordination polymer **1** can be treated as 4-connected **sql** type uninodal net with point symbol {4^4^.6^2^}.

### 2.3. Powder X-ray Diffraction, IR Spectroscopy, and Thermal Behavior

PXRD analysis was performed to confirm the phase purity of **1**, its stability toward solvent, and metal inclusion. Comparison of the experimental and calculated diffraction patterns (Figure 3) confirms that the bulk product is a pure single phase, that is quite stable on prolonged exposure to moist air and DMA.

The thermal stability was investigated by means of thermogravimetric (TGA) analysis. The obtained TG curve suggests that **1** is thermally stable to approximately 320 °C. There is a clear step on the curve (Appendix A) that represents solvent loss (18% observed mass loss, calculated 18.3% for three DMF, and 3.76 EtOH molecules per elementary unit, which is consistent with elemental analysis and X-ray crystal structure data). Above 320 °C, degradation of the framework is clearly observed, which reaches completion at 730 °C. Overall, coordination polymer **1** demonstrates a relatively good thermal stability.

The IR spectrum (Appendix A) contains several characteristic bands. Strong bands located at 1656 and 1386 cm^−1^ can be attributed to the asymmetric and symmetric carboxylate stretching vibrations. The separation between these bands Δ = 270 cm^−1^ falls within the range of values usually observed for the asymmetric bidentate carboxylate coordination mode [34], which is consistent with the crystal structure data. Bands in the region of 1100–900 cm^−1^, along with a sharp band at 1527 cm^−1^ can be attributed to 1,2,4-triazole ring vibrations [35].

### 2.4. Luminescent Properties

Luminescent behavior of the coordination polymer **1** was studied both in the solid state and in DMA suspension. Compound **1** exhibits bright turquoise luminescence both in solid state and in DMA suspension. The emission maximum of **1** (λ_em_ = 490 nm, λ_ex_ = 375 nm, Figure 4) is slightly blue-shifted compared to the free tr_2_btd ligand (λ_em_ = 510 nm, λ_ex_ = 375 nm, Appendix A), which indicates that not only tr_2_btd, but also bpdc^2−^ ligand (λ_em_ = 480 nm, λ_ex_ = 375 nm, Appendix A) are responsible for the luminescence of the framework. Moreover, there are three distinct bands in the excitation spectrum of **1** in DMA suspension, with the maxima located at 282 nm, 320 nm, and 375 nm (Figure 4b), which further proves that both ligands participate in the excitation process. However, since there are no additional bands in the emission spectrum of **1** and in the excitation spectrum of the free ligand, a ligand-to-ligand charge transfer process can be assumed. Interestingly, a red shift could be noted when **1** is soaked in DMA, and emission maximum can be observed at 507 nm (λ_ex_ = 375 nm, Figure 4b).

### 2.5. Metal Ion Sensing

For the metal ion sensing experiments, DMA was chosen as a dispersion medium. For the screening test, metal salt solutions were added to the suspension to achieve concentrations of 0.01 mM. The results of the screening tests are shown in Figure 5a. It is quite obvious that the majority of metal ions (Zn^2+^, Cd^2+^, Ni^2+^, Co^2+^, Cu^2+^, Eu^3+^, La^3+^, Ba^2+^, Sr^2+^, Mg^2+^, K^+^, Na^+^, Pb^2+^) had negligible quenching effect on the luminescence intensity. In case of Cr^3+^ and Fe^3+^ a slight (1.1 intensity of the blank sample) increase was observed. Aluminum undeniably stands out, as concentration of 0.01 mM causes significant luminescence enhancement leading to a two-fold increase in the intensity (Figure 5a).

In addition, concurrent sensing experiments were performed, when equal concentrations of Al^3+^ and one of the other metal cations (Zn^2+^, Cd^2+^, Ni^2+^, Co^2+^, Cu^2+^, Eu^3+^, La^3+^, Ba^2+^, Sr^2+^, Mg^2+^, K^+^, Na^+^, Pb^2+^) were introduced into the suspension. It was found that the presence of various cations did not interfere with the ability of **1** to detect aluminum, confirming the excellent selectivity of its luminescent response toward Al^3+^ ion (Figure 5b).

Photoluminescence quantum yields (QY) and luminescence lifetimes of the pristine suspension and suspension with 0.005 mM Al^3+^ were measured. The luminescence decay of **1** could be described by a two-exponential model with the lifetimes (4.8 ns and 16.6 ns, Appendix A) characteristic for ligand-centered fluorescence. Upon addition of Al^3+^, the lifetimes did not change significantly (2.6 ns and 16.1 ns, Appendix A) suggesting a static fluorescence enhancement mechanism with the formation of an emissive complex in the ground state [36]. Upon addition of Al^3+^, a change in luminescence quantum yield from 24% to 40% was observed. To further explore the relationship between Al^3+^ concentration and luminescence properties of **1**, fluorimetric titration experiments with increasing concentration of Al^3+^ in the suspension were carried out (Figure 6). Two linear regions (for concentration ranges 0.5–3 μM and 4–8 μM) could be observed in the relationship between integrated fluorescence emission intensity ratios (I/I_0_) and Al^3+^ concentration (Figure 6b and Appendix A). The detection limit was calculated from the lower concentration approximation using the well-established equation 3 σ/k (where σ is standard deviation calculated from 5 blank measurements and k is slope of the graph) [37] to be 120 nM. This value is among the best reported Al-sensing MOF materials (Table 1). These results indicate that **1** can be used as an effective, sensitive, and selective fluorescent probe for Al^3+^.

Reusability is an important factor in any practical application. To explore the possibility of using **1** for the detection of Al^3+^ several consecutive times, **1** was soaked in Al^3+^ solution, then the fluorescence spectrum was recorded, the framework was filtered out, washed with ethanol, and soaked again. We ran the sample through additional three cycles, and the results are shown in Appendix A. It is evident that both intensity and the luminescent response of **1** did not suffer significant change on repeated sensing experiments.

To demonstrate that **1** can be used for practical application in the detection of Al^3+^ ions, fluorescent paper strips were prepared by sonicating strips of filter paper in the DMA suspension of **1**. The paper was dipped in the 0.01 mM solution of Al(NO_3_)_3_ in DMA. As it is evident from the photo (Figure 7), **1** can be used for the practical naked-eye detection of Al^3+^ present in the solution.

To further explore the applicability of 1 for Al^3+^ determination in real samples, tap water was analyzed using the calibration regressions described above. For this, 100 mL of tap water was evaporated to dryness, 1 mL of DMA was stirred with the solid residue for 20 min, the insoluble residue was filtered out, and 100 μL of the filtered DMA were introduced to the **1** suspension. The emission spectra for the blank and tap water samples were recorded and compared, and Al^3+^ concentration in the tap water was calculated to be approximately 0.2 μM. To elucidate the possible influence of the analytical matrix (i.e., all other metal cations and anions present in the tap water), 120 μL of 10^−4^ M Al^3+^ solution was added to both samples (total volume was 2 mL). After leaving the samples to equilibrate for 30 min, emission spectra were recorded again (Appendix A) and aluminum recovery between spiked real and blank samples was calculated to be 91%. The experiment was repeated three times and the method precision (as relative standard deviation) was calculated to be 3.7% and an accuracy of 4.8% was achieved.

To date, several possible mechanisms responsible for the luminescence enhancement upon guest inclusion into the coordination polymer were described: (i) Structural reorganization of the framework; (ii) exchange of metal cations; (iii) specific non-covalent interactions with the pore or channel surface. Possible structural changes were unequivocally ruled out, as PXRD pattern of **1** soaked in Al^3+^ solution is consistent with the as-synthesized and calculated patterns (Figure 3). As **1** is a neutral framework it is highly unlikely that any cation exchange takes place within the structure.

Time-dependent fluorescence measurements afford some insight into the mechanistic side of the observed phenomenon, as after the addition of Al^3+^ fluorescence intensity slowly rises, reaching a maximum after approximately 10 min (Appendix A). This may be due to kinetics of Al^3+^ ions permeation into the framework channels and windows. CIE chromaticity diagram reveals a slight change in luminescence color (Appendix A), which shifts into the green area of the spectrum, which can be a sign of Al^3+^ ions interacting with tr_2_btd molecules in the structure, hindering the mobility of the fragments, thus reducing the amount of energy released through non-radiative pathways. This leads to the increase of tr_2_btd contribution in the emission of the framework, which results in intensity increase and slight color shift (tr_2_btd emission maximum is located at 515 nm). Aluminum nitrate in DMA (in the concentration range used in this study) does not show any absorption in the excitation region of 1, thus energy transfer leading to luminescence quenching can be ruled out.

Overall, when compared to the other reported Al^3+^ sensitive coordination polymer-based materials (Table 1), **1** exhibits excellent performance with regard to sensitivity and selectivity and has a significant potential to be used as a fluorescent probe for Al^3+^.

## 3. Materials and Methods

All chemicals except for tr_2_btd were purchased from commercial sources, were of at least analytical grade, and used without additional purification.

NMR spectra were recorded with a Bruker Advance III (500 MHz) instrument.

IR spectra in KBr pellets were recorded in the range 4000−400 cm^−1^ on a Bruker Scimitar FTS 2000 spectrometer.

Thermogravimetric analysis was done on a NETZSCH TG 209 F1 thermobalance in a stream of He in the range of 50 to 800 °C, heating rate 10 °C/min.

Photoluminescence spectra and luminescence decay kinetics were recorded on Horiba Fluorolog 3 (HORIBA Jobin Yvon SAS, Edison, NJ, USA) equipped with 450W ozone-free Xe lamp (HORIBA Jobin Yvon SAS, Edison, NJ, USA), a cooled PC177CE-010 photon detection module (HORIBA Jobin Yvon SAS, Edison, NJ, USA) with R2658 photomultiplier, and double grating excitation and emission monochromators. Absolute quantum yields were determined using Quanta-ϕ integrating sphere (HORIBA Jobin Yvon SAS, Edison, NJ, USA). Excitation and emission spectra were corrected for source intensity (lamp and grating) and emission spectral response (detector and grating) by standard correction curves. For measurements powdered samples were placed between two non-fluorescent quartz plates. Photoluminescence measurements of DMA suspensions were carried out in 1 cm quartz cuvettes.

Powder X-ray diffraction (PXRD) analysis was performed at room temperature on a Shimadzu XRD-7000 diffractometer (Shimadzu Corporation, Kyoto, Japan, Cu-Kα radiation, λ = 1.54178 Å).

Elemental analysis was carried out on a Vario Microcube analyzer (Elementar Analysensysteme GmbH, Langenselbold, Germany).

### 3.1. Synthesis of 4,7-di(1H-1,2,4-triazol-1-yl)-2,1,3-benzothiadiazole

Anhydrous K_3_PO_4_ (3.61 g, 17 mmol) was suspended with 1H-1,2,4-triazole (1.17 g, 17 mmol) in 50 mL of dimethyl sulfoxide and heated at 80 °C with stirring for 30 min. 4,7Dibromobenzo-2,1,3-thiadiazole (1.00 g, 3.4 mmol) was added and the mixture was stirred at 110 °C for 24 h. Then the mixture was poured on ice-water and filtered. Greenish-brown crude product was drier on air and pure product was obtained by recrystallization from DMF:EtOH (1:1) mixture. Bright-green needles were filtered from the red mother liquor and dried on air at 60 °C. Yield 432 mg (47%), green needles. ^1^HNMR (500 MHz, CDCl_3_): δ 8.21 (s, 2H, btd), 8.43 (s, 2H, 3-H-Tr), 9.80 (s, 2H, 5-H-Tr) ppm. ^13^C¬NMR (125 MHz, CDCl_3_): δ 123.8 (4,7-C-btd), 129.3 (5,6-C-btd), 141.4 (3-C-Tr), 150.4 (4a,7a-C-btd), 196.3 (5-C-Tr). NMR data (Appendix A) are consistent with the literature data [32].

### 3.2. Synthesis of [Zn(tr_2_btd)(bpdc)]∙DMF *(**1**)*

Total of 5 mg (0.018 mmol) of tr_2_btd, 4.5 mg (0.018 mmol) of biphenyl dicarboxylic acid, and 5.5 mg (0.018 mmol) of zinc nitrate nitrate hexahydrate were dissolved in 1 mL of DMF:EtOH (3:1) mixed solvent with sonication and heated at 100 °C for 24 h. Yellow-greenish needle crystals were filtered out, single crystals were picked out for the X-ray analysis. Yield 78%. C_27_H_21_N_9_O_5_SZn∙1.25C_2_H_5_OH (706.55): calcd. C 50.00; H 4.31; N 17.85; S 4.54; found: C 49.6; H 3.5; N 17.9; S 4.7. FT-IR (cm^−1^): 3500 (m), 3000 (m), 1656 (s), 1608 (s), 1527 (s), 1386 (s), 1282 (m), 1112 (s), 983 (s), 933 (w), 858 (s), 773 (s), 671 (s), 615 (w), 439 (w).

### 3.3. Luminescence Sensing Experiments

Suspension of coordination polymer for screening test in DMA was prepared as follows: 10 mg of **1** was carefully ground in mortar and then sonicated for 1 h in 10 mL of DMA and left standing for 24 h. The supernatant (stable suspension) was separated and used in luminescence sensing experiments.

For the screening test, 250 µL of suspension was diluted to 2 mL with fresh DMA and metal salt solution was added to achieve a concentration of 0.01 mM.

### 3.4. Single Crystal X-ray Analysis

Single crystal XRD data for **1** were collected with a Bruker D8 Venture diffractometer with a CMOS PHOTON III detector and IµS 3.0 source (Mo Kα radiation, λ = 0.71069 Å). All measurements were conducted at 150 K, the φ- and ω-scan techniques were employed. Absorption corrections were applied with the use of the SADABS program [46].

The structure was solved by the direct methods and refined by the full-matrix least squares methods using SHELXT and SHELXL [47] with SHELXle GUI [48]. The non-hydrogen atoms were situated in successive difference Fourier syntheses and refined by anisotropic thermal parameters on F2. Hydrogen atoms were placed geometrically and treated as a mixture of independent and constrained refinement. The disordered electron density, which could not be satisfactorily modeled into solvent molecules was removed with the aid of PLATON SQUEEZE algorithm (electron count removed was 173, which corresponds to 3.76 molecules of EtOH per formula unit).

Crystal data for C_27_H_21_N_9_O_5_SZn (*M* = 648.98 g/mol): monoclinic, space group *P*2_1_/*c*, *a* = 13.3256(3) Å, *b* = 13.3256(3) Å, *c* = 13.3256(3) Å, β = 104.9180(10)°, *V*= 8470.5(4) Å^3^, Z = 12, T = 150(2) K, μ(MoK_α_) = 0.12 mm^−1^, *D_calc_* = 1.527 g/cm^3^, 107669 reflections measured, 26362 unique (*R*_int_ = 0.093, *R*_sigma_ = 0.056). The final *R*_1_ was 0.0563 (I > 2σ(I)), goodness of fit S was 1.027 and *wR*_2_ was 0.141 (all data).

## 4. Conclusions

In summary, by exploiting the mixed-ligand design approach, a new 2D coordination polymer, which can be utilized as a “turn-on” fluorescence probe for the Al^3+^ in solution was prepared. The material obtained demonstrated suitable chemical stability and excellent sensitivity, selectivity for the presence of Al^3+^. The detection limit for Al^3+^ was among the lowest reported for the Al-sensitive coordination polymers. These results demonstrate the significance of a judicious ligand choice in the design of functional materials with desirable properties.

## Data Availability

CCDC 2121288 contains the Appendix A for this paper. These data can be obtained free of charge from The Cambridge Crystallographic Data Center at http://www.ccdc.cam.ac.uk/data_request/cif (accessed on 5 December 2021).

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
