# Peer review of "A Zn(II)-Based Sql Type 2D Coordination Polymer as a Highly Sensitive and Selective Turn-On Fluorescent Probe for Al3+"

_molecules, 2021, doi:10.3390/molecules26237392_

Round 1

Reviewer 1 Report

Review Report

This paper reports MOF-based sensing of Al3+ ions relying on fluorescence enhancement. Overall, this manuscript is well-written and properly organized. However, some concerns should be addressed before its acceptance.

  1. Is chromatography a conventional method for the detection of aluminum? Please verify.
  2. Add decimal instead of a comma in the values provided on the x and y-axis of figures 4, 5, and 6.
  3. Some elements such as Pb and Na significantly decrease the luminescence intensity of Aluminum ions? How would you explain this?
  4. It would have been more useful if the calibration had been developed in the presence of other metal ions.
  5. Why was this sensing not used for the detection of Al3+ in any real samples? I suggest using this probe for some real environmental or biological samples. Otherwise, there is no practical relevance to developing such a sensing platform.
  6. The plots should be an average of triplicate experiments and should include error bars.
  7. Please include all the analytical figures of merits, such as precision ( %RSDs) and accuracy.
  8. Moreover, real samples should be spiked to prove that recoveries of Al3+ are in the acceptable range.

Author Response

Thank you very much for evaluating our work and pointing out to important analytical aspects. Here are our replies to the specific comments.

This paper reports MOF-based sensing of Al3+ ions relying on fluorescence enhancement. Overall, this manuscript is well-written and properly organized. However, some concerns should be addressed before its acceptance.

1. Is chromatography a conventional method for the detection of aluminum? Please verify.

Ion chromatography can indeed used for Al determination. Specific literature references for each of the listed methods were added (lines 50-52).

2. Add decimal instead of a comma in the values provided on the x and y-axis of figures 4, 5, and 6.

Corrected.

3. Some elements such as Pb and Na significantly decrease the luminescence intensity of Aluminum ions? How would you explain this?

When the experiments were carefully repeated in triplicate, Na+ and Pb2+ did not significantly influence the analytical signal for Al and almost a two-fold intensity increase was observed. We believe the initial decrease was caused by random errors, which were eliminated in repeated experiments. We are thankful to the reviewer for pointing the deviation for Pb and Na.

4. It would have been more useful if the calibration had been developed in the presence of other metal ions.

When studying the luminescent/spectrophotometric response of different systems to metal ions, it is generally accepted to obtain calibration curves for solutions of pure substances. In real life applications, in each specific case, it is possible to build calibration for a specific set of interfering ions/components. The major aim of the work was to report a new coordination polymer and demonstrate its potential to detect aluminum ions, thus we follow a general methodology and present a calibration for pure analyte (Al3+).

5. Why was this sensing not used for the detection of Al3+ in any real samples? I suggest using this probe for some real environmental or biological samples. Otherwise, there is no practical relevance to developing such a sensing platform.

The determination procedure of Al3+ in tap water was added to the text (lines 206-218). We should also point out that development of a complete analytical procedure was outside the scope of this synthetic work, but may be an object of a separate work, perhaps including several luminescent MOFs.

6. The plots should be an average of triplicate experiments and should include error bars.

Triplicate experiments were carried out, the error bars were added to the plots

7. Please include all the analytical figures of merits, such as precision ( %RSDs) and accuracy.

Accuracy and precision values for Al determination in tap water matrix were added (lines 216-218).

8. Moreover, real samples should be spiked to prove that recoveries of Al3+ are in the acceptable range.

Spiking experiment to tap water matrix was carried out, the obtained recovery was added to the text (line 216).

Reviewer 2 Report

In this manuscript, the authors report a 2D zinc coordination polymer that features a “turn-on” fluorescence in contact with Al3+ ion. The polymer is characterized by crystallography, XRD and IR spectroscopy and shows good thermal stability. The fluorescence of the polymer is increased when mixed with Al3+ and shows selectivity over other metal cations. The sensing of Al3+ is highly sensitive and the material can be reused multiple times. Overall, this work provides a new method for selective Al3+ sensing with high sensitivity.

Comments:

  • It will be friendly for the audience to add the structure of the polymer 1 in Fig. 1 when it is first call-out on line 67.
  • The authors state that the syntheses of ligand tr2btd and polymer 1 afford moderate and high yield. The authors are encouraged to report the actual isolation yield for each reaction, especially for tr2btd since they claim that the recrystallization method presents advantages over chromatography.
  • In section 2.4, the authors state that the excitation wavelength for the free tr2btd ligand is 490 nm. However, in the SI figure S5, it shows the excitation wavelength is 350 nm. The authors are encouraged to explain the difference.
  • For section 2.5, the authors need to leverage more on the luminescent experiments in:
    1. The authors only report the excitation spectra of the polymer and the ligand and claimed that the fluorescence of the polymer results from both ligands. To support this claim, the comparison between the electronic absorption spectrum and the excitation spectrum must be made.
    2. The absorption, excitation, and emission spectra of the bpdc2- ligand are missing. To support the above-mentioned conclusion, these data are needed.
    3. The authors claim that the red-shifted emission of the polymer 1 in DMA is due to low transmittance at 270 nm thus reduces the energy absorbed by bpdc2- ligand. I don’t understand this because the excitation wavelength for the emission experiment is at 490 nm, anything absorbs at 270 nm will not be excited anyway.
  • On line 174, the authors mention that the QY do not change much with addition of Al3+ because internal absorption of the suspension. If the authors mean reabsorption during the emission experiments, it could be possible that the QY determination is not accurate. The concentration of the fluorophore may be too high for the QY determination. Experiments with lower concentrations are suggested to rule out the reabsorption issue.
  • The absorption data with the Al3+ sensing experiments are missing from the manuscript, and they should be included before draw conclusions on this study.
  • They authors claim the theoretical detection limit in the manuscript multiple times and compare with other’s detection limit data. However, experimental data for detection only reach 17 microM that is 220 folds greater than the 77 nM (0.077 microM) claimed in this work. While reaching to 0.1 microM of the metal ion solution is not hard to achieve, the authors are encouraged to present lower experimental detection limit.

In summary, this manuscript needs a major revision and reconsideration before it can be recommended for publication in Molecules.

Author Response

In this manuscript, the authors report a 2D zinc coordination polymer that features a “turn-on” fluorescence in contact with Al3+ ion. The polymer is characterized by crystallography, XRD and IR spectroscopy and shows good thermal stability. The fluorescence of the polymer is increased when mixed with Al3+ and shows selectivity over other metal cations. The sensing of Al3+ is highly sensitive and the material can be reused multiple times. Overall, this work provides a new method for selective Al3+ sensing with high sensitivity.

Comments:

  • It will be friendly for the audience to add the structure of the polymer 1 in Fig. 1 when it is first call-out on line 67.

The structure of the coordination polymer 1 is complicated to show on one figure with the structures of the ligands. In the first mention of polymer 1 we only show its chemical composition and introduce ligand abbreviations (shown in Fig. 1). The crystal structure of compound 1 is discussed later in section 2.2. To preserve this line of thought we prefer to separate the structural formula of the ligands (Fig. 1) from the crystal structure representation of the coordination polymer 1 (Fig. 2).

  • The authors state that the syntheses of ligand tr2btd and polymer 1 afford moderate and high yield. The authors are encouraged to report the actual isolation yield for each reaction, especially for tr2btd since they claim that the recrystallization method presents advantages over chromatography.

The isolated yields of tr2btd and polymer 1 are given in the corresponding synthetic procedures in the Experimental part.

  • In section 2.4, the authors state that the excitation wavelength for the free tr2btd ligand is 490 nm. However, in the SI figure S5, it shows the excitation wavelength is 350 nm. The authors are encouraged to explain the difference.

The maximum excitation wavelength for tr2btd is 375 nm (404 nm was a typo) as is stated in section 2.4 (490 nm is the emission maximum for compound 1). Since the emission maximum of tr2btd is independent on the excitation wavelength, the emission spectrum in Fig. S5 is recorded at shorter excitation wavelength to obtain better signal-to-noise ratio. To avoid misunderstanding, the figure was updated to show the maximum excitation wavelength of 375 nm.

  • For section 2.5, the authors need to leverage more on the luminescent experiments in:

1. The authors only report the excitation spectra of the polymer and the ligand and claimed that the fluorescence of the polymer results from both ligands. To support this claim, the comparison between the electronic absorption spectrum and the excitation spectrum must be made.

Since we only discuss the luminescence spectra of powdered samples or fine suspensions of the coordination polymer (which are solid state structures by definition), the absorption spectra of the ligands in solution seem to be not relevant for solid-state excitation/ absorption.

2. The absorption, excitation, and emission spectra of the bpdc2- ligand are missing. To support the above-mentioned conclusion, these data are needed.

The spectra of H2bpdc ligand were added (Figure S6).

3. The authors claim that the red-shifted emission of the polymer 1 in DMA is due to low transmittance at 270 nm thus reduces the energy absorbed by bpdc2- ligand. I don’t understand this because the excitation wavelength for the emission experiment is at 490 nm, anything absorbs at 270 nm will not be excited anyway.

This statement was deleted from the manuscript, thank you for pointing out the discrepancy.

  • On line 174, the authors mention that the QY do not change much with addition of Al3+ because internal absorption of the suspension. If the authors mean reabsorption during the emission experiments, it could be possible that the QY determination is not accurate. The concentration of the fluorophore may be too high for the QY determination. Experiments with lower concentrations are suggested to rule out the reabsorption issue.

The quantum yields were re-measured with more diluted suspensions, the same suspensions were also used for titration experiments. (lines 175-178).

  • The absorption data with the Al3+ sensing experiments are missing from the manuscript, and they should be included before draw conclusions on this study.

Al does not show any absorption in the excitation region of the MOF (375 nm). If there was an overlap between the absorption and excitation bands, luminescence quenching would be observed, which is not the case for our system. A note was added in lines 235-238.

  • They authors claim the theoretical detection limit in the manuscript multiple times and compare with other’s detection limit data. However, experimental data for detection only reach 17 microM that is 220 folds greater than the 77 nM (0.077 microM) claimed in this work. While reaching to 0.1 microM of the metal ion solution is not hard to achieve, the authors are encouraged to present lower experimental detection limit.

When the experiments with the extended concentration range were repeated, two linear regions were found, the plots were added to the manuscript (Figure 6b and S9), the LOD was determined from the lower-concentration regression and the refined value is 120 nM.

Round 2

Reviewer 1 Report

The authors have tried to address my comments, although I am not agree with their opinion that this synthetic work does not need full demonstration of analytical methodology. 

Reviewer 2 Report

The revised manuscript is recommended for publication as a full article in Molecules.